# Revisiting the Woolly wolf (*Canis lupus chanco*) phylogeny in Himalaya: Addressing taxonomy, spatial extent and distribution of an ancient lineage in Asia

**BheemDutt Joshi**[1]*, **Salvador Lyngdoh**[1]*, **Sujeet Kumar Singh**[1], **Reeta Sharma**[1], **Vinay Kumar**[1], **Ved Prakash Tiwari**[1], **S. A. Dar**[1], **Aishwarya Maheswari**[1], **Ranjana Pal**[1], **Tawqir Bashir**[1¤], **Hussain Saifee Reshamwala**[1], **Shivam Shrotriya**[1], **S. Sathyakumar**[1], **Bilal Habib**[1]*, **Laura Kvist**[2], **Surendra Prakash Goyal**[1]*

**1** Wildlife Institute of India, Chandrabani, Dehradun, India, **2** Department of Biology, University of Oulu, Oulu, Finland

¤ Current address: Centre of Research for Development, University of Kashmir, Srinagar, India
* goyalsp@wii.gov.in (SPG); salvador@wii.gov.in (SL); bh@wii.gov.in (BH); joshidutt01@gmail.com (BJ)

## Abstract

Of the sub-species of Holarctic wolf, the Woolly wolf (*Canis lupus chanco*) is uniquely adapted to atmospheric hypoxia and widely distributed across the Himalaya, Qinghai Tibetan Plateau (QTP) and Mongolia. Taxonomic ambiguity still exists for this sub-species because of complex evolutionary history anduse of limited wild samples across its range in Himalaya. We document for the first time population genetic structure and taxonomic affinity of the wolves across western and eastern Himalayan regions from samples collected from the wild (n = 19) using mitochondrial control region (225bp). We found two haplotypes in our data, one widely distributed in the Himalaya that was shared with QTP and the other confined to Himachal Pradesh and Uttarakhand in the western Himalaya, India. After combining our data withpublished sequences (n = 83), we observed 15 haplotypes. Some of these were shared among different locations from India to QTP and a few were private to geographic locations. A phylogenetic tree indicated that Woolly wolves from India, Nepal, QTP and Mongolia are basal to other wolves with shallow divergence (K2P; 0.000–0.044) and high bootstrap values. Demographic analyses based on mismatch distribution and Bayesian skyline plots (BSP) suggested a stable population over a long time (~million years) with signs of recent declines. Regional dominance of private haplotypes across its distribution range may indicate allopatric divergence. This may be due to differences in habitat characteristics, availability of different wild prey species and differential deglaciation within the range of the Woolly wolf during historic time. Presence of basal and shallow divergence within-clade along with unique ecological requirements and adaptation to hypoxia, the Woolly wolf of Himalaya, QTP, and Mongolian regions may be considered as a distinct an Evolutionary Significant Unit (ESU). Identifying management units (MUs) is needed within its distribution range using harmonized multiple genetic data for effective conservation planning.

**Data Availability Statement:** All relevant data are within the paper and its Supporting Information files.

**Funding:** Wildlife Institute of India Grant in aid funds.

**Competing interests:** The authors have declared that no competing interests exist.

# Introduction

Resolving species and sub-species ambiguity has been a key issue in conservation biology, especially in widely distributed and species having the large home ranges [1]. Such ambiguity of classification into species or sub-species has often been debated in the literature for extant mammalsof the Central Asian region as well. Some examples of the debated species in Qinghai Tibetan Plateau (QTP) and Himalayan regions are musk deer (*Moschus* spp.) [2], blue sheep [3], wolf [4], brown bear [5]and snow leopard [6,7]. This region is known to have undergone various climatic changes in the past and differences in uplift since the Pleistocene. These events may have even lead to distinctpaleo-ecological niches because of differences in major geophysical events in these two (QTP and Himalaya) regions[3] and could have resulted in distinct evolutionary significant units (ESU)/management units.

Of the medium sized carnivores of the world, the grey wolf (*Canis lupus*) is one of the most widely distributed species in different bioclimatic zones and it is divided into many discrete populations and subspecies across the world [8]. Several studies have addressed the issues of phylogeography, genetic diversity, taxonomy and evolutionary history in the northern hemisphere[4,9–11]. However, the Woolly wolf (*Canis lupus chanco*) of the Qinghai Tibetan Plateau-Himalaya region has not caught much attention. It is one of the sub-species of Holarctic wolf (*Canis lupus)* and is widely distributed in Central Asia ranging from Chinese Turkestan and the Tian Shan throughout Tibet to Mongolia, North China, Shensi, Szechwan, Yunnan and the western Himalaya [12]. This sub-species is also known by different names across its range. Several types for different localities have been reported, such as "*chanco*" in Chinese Tartary; "*laniger*" in Tibet; "*niger*" near Hanle in S.E. Kashmir; "*filchneri*" in Si-nung-fu, Kansu; "*karanorensis*" in Kara-nor, the Gobi; and *tschiliensis*" in the coast of Chihli [12].

The taxonomic status of the Himalayan and peninsular wolves (wolf distributed in peninsular India)has been revised several times, based on the ecological and morphological studies, and accorded as a sub-species ora species. The Woolly wolf from the Himalayan region was designated as distinct species called *C. laniger* [13]which later on was merged with the *C. lupus*[14], whereas the lowland Indian wolf was assigned as a species named *C. pallipes* Blanford (1888) [14]. Further, these both were designated as sub-species of *C. lupus* and named as *C. l. chanco* and *C.l. pallipes* respectively.Subsequently, it was also agreed on the taxonomic classification of the Himalayan wolf[15][16]that was proposed as *C. l. chanco*[13]and Peninsular wolf as *C. l. pallipes*[14]. Currently, the taxonomic classification of the Woolly wolf is still under debate [16] and is pending due to limited data from contemporary wild population across Himalaya.

The limited studies undertaken on the Woolly wolf from Himalaya, QTP and Mongolia have assigned it at a basal position in an evolutionary cladogram and considered it as an ancient lineage [10,15,17]. Later on samples analysed [15]from Nepal, Tibet, Jammu and Kashmir and mentioned that wolves from the Himalayan region formeda separate clade[15]. They further suggested that the Pleistocene geological events may have shaped the observed phylogeographic pattern. Subsequently, the taxonomic status of the wolf of the Indian sub-continent was examined using mitochondrial control region sequences[18] and suggested new species' named *C. himalayensis* for the Himalayan wolf and *C. indica* for the Peninsular Indian wolf. However, most of these studies have relied mostly on limited samples that originated from either zoos or museums and, for example, the relatedness of the zoo individuals was not studied [16]. A recent study using wild-caught samples of the Woolly wolf of the Himalayan region from Nepal suggested phylogenetic distinction and old divergence from the Holarctic wolf [11]. Likewise, Werhahn et al. [4,19]suggested distinctiveness of the Woolly wolf from other wolves and advocated that their geographic range is not restricted to the Himalaya. They also recommended that the Woolly wolf should be classified as a distinct taxon of special

conservation concern. However, all these studies have used limited samples across distribution range of wild wolves from the Himalayan region.

In the current study, we re-examine the phylogeny and population genetic structure of the Woolly wolf based on a wide coverage of samples collected from the wild across the Himalayan region of India and compare the data with the published literature available for the Himalayan region of Nepal, QTP, and Mongolia. We also examine the possible extent of the Woolly wolf lineage relationship within Holarctic grey wolf [4,19].

# Material and methods

## Ethical statement

All the samples were non-invasively collected from the different states of Indian Himalayan region. Fecal samples of wolf were collected noninvasively without any animal capture or handling. Therefore, sample collection did not require any handling permission from the respective department. However, permission for procuring, processing and preserving scat samples were obtained from the from the Ministry of Forest, Environment and Climate Change, Uttarakhand with letter vide no 1/29/2003-PT, Department of Forest, Environment and Wildlife Management, Government of Sikkim vide No. 2081778 Dated 18/04/2012, Department of Forest and Environment and Climate Change of Himachal Pradesh, vide no. WL/Research Study/WLM/4671 dated 24/11/2015 and WL/Research Study/WLM 87 dated 07/04/2010.

## Sample collection and DNA extraction

We collected putative wild wolf scat samples (n = 132) from Indian Himalayan range covering four regions of Jammu and Kashmir (JK) (n = 5), Himachal Pradesh (HP) (n = 120), Uttarakhand (UK) (n = 4) and Sikkim (SK) (n = 3) during 2003–2015 with an elevation ranges from 3000 to 4500 m (Fig 1). DNA was extracted from the samples using the QiagenDN easy Stool Kit, Qiagen, Germany, according to the manufacturer's instructions. To detect any contamination in the DNA extraction procedure and reagents, we performed the extraction procedure with two independent negative controls.

## Selection of primer

Various molecular markers such as microsatellites, mitochondrial regions, whole-genome sequencing (WGS), single nucleotide polymorphism (SNPs) have been suggested for identifying management units (MU), Evolutionary Significant Units (ESU) and Distinct Population Segment (DPS) for effective conservation planning across species' range [20–22]. With the advances in molecular genomics, WGS has been a preferred a genomic approach in conservation studies but requires good quality of samples [23–25]. Obtaining adequate number of invasive samples such as blood or tissue for WGS studies from the widely distributed species of low density are not cost-effective in comparison to fecal DNA [26,27] and also lacks comparable data at present to document variations due to microevolution. Woolly wolf have several "Ecomorph" and obtaining adequate invasive samples across its range would be a challenge for planning WGS studies. Besides, bringing compatible harmonized microsatellite data for defining conservation units has also been a challenge for species having transboundary range [28]. However, the control region (CR) of the mitochondrial genome 'is the highly variable'region and 250 to 450 bp have been used in several phylogenetic/phylogeographic studies on wolves [4,29,30]. Therefore, we preferred to use CR region of 250 bp in the present study, which is easy to amplify with faecal DNA. This covers maximum variability and extensive comparable

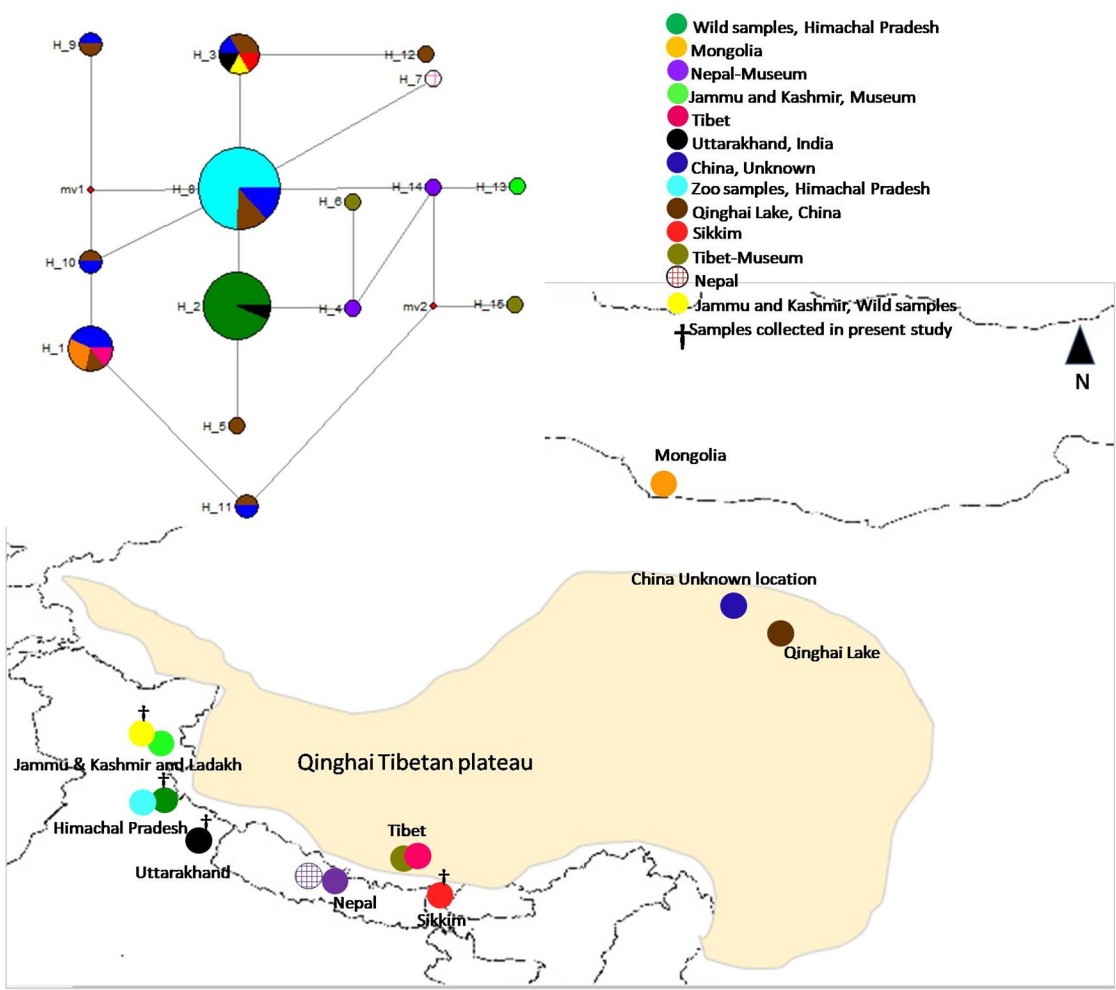

**Fig 1. Median Joining network of different haplotypes of *Canis lupus chanco* from different geographical regions.** Circle sizes are proportional to the number of samples with that haplotype using the mitochondrial control regions. Circle on the map represent the sampling locations (map not to scale).

data are available from almost throughout the range, including the study of Nepal and others [11,15,19,31].

**PCR amplification and DNA sequencing.**  Species of each scat was identified using mitochondrial cytochrome b gene with primers widely used in carnivores species [32]. Cytochrome b gene was amplified in 20 μl PCR, containing 2.0 μl DNA template, 1.6 mM MgCl2, 2x Buffer, 0.5 unit Tag DNA polymerase, 2.4 μldNTPs and 1.0 μM each primer. PCR cycling was performed with an initial denaturation for 5 minutes at 94 ˚C, followed by 40 cycles of denaturation for 45 s at 94 ˚C annealing for 60 s at 55 ˚C and 1 min extension at 72 ˚C with a final extension of 72 ˚C for 10 min. Only scats confirmed to be of *Canis lupus* (n = 19) were used for further analysis. These were from Jammu and Kashmir (n = 1), Himachal Pradesh (n = 15), Uttarakhand (n = 2) and Sikkim (n = 1). Mitochondrial control region (CR) was amplified with primers in [33] as above, except the annealing temperature was 56 ˚C for 45 s. The cycle sequencing of PCR products were cleaned using the BigDye Terminator kit (Applied Biosystems, USA) and purified products were subjected to DNA sequencing on the ABI 3730 Genetic Analyzer (Applied Biosystems, USA).

## Data analysis

DNA sequences of cytochrome b and control region were examined and edited, and aligned using CLUSTAL W [34] as implemented in BioEdit v.7.2.5[35]. All electropherograms were inspected by eye for quality, especially for the observed variable sites and validated using Sequencher 4.7 (Gene Codes Corporation, USA). Generated sequences were also validated using reference data from the NCBI, BLAST at GenBank (http://www.ncbi.nlm.nih.gov). Besides sequences generated in this study, we also retrieved total 66 sequences of *Canis lupus* sub-species control region from the GenBank originating from Peoples Republic of China (PRC), Mongolia, Japan and India, and which were defined as *C.l. chanco*. We also retrieved one sequence that was assigned as *Canis lupus laniger* from Tibet. List of sequences used in this study are provided in the electronic supplementary file in S1 Table. All further analysis were done with control region data only.

Divergence of sequences was calculated using the Kimura 2 parameters (K2P)distance matrix in the MEGA 6[36]. Genetic diversity was estimated by computing nucleotide ($\pi$) and haplotype (h) diversity in DnaSP 5.10 [37] which was used for neutrality tests (Tajima D and Fu's *Fs*) as well. Nucleotide diversity calculated as the average number of nucleotide differences per site whereas haplotype diversity is the probability how two randomly sampled haplotypes are different[38]. Tajima D statistic used to measure the observed variation of a population with respect to variation expected in a randomly evolving population [39]. Fu's Fs statistics is used to point the discrepancy between nucleotide differences observed in the samples[40]. Population differentiation ($F_{ST}$) as a measure of structure in natural population, ruggedness indices and mismatch distributions, implying size changes or stationarity of populations, were calculated using Arlequin v. 3.1.1[41]. The models for nucleotide substitution were selected comparing Akaike Information Criterion 2 scores implemented in the Model Generator [42]Patterns of historical demography can also be inferred from estimates of the effective population size over time using theBayesian skyline plot(BSP) method implemented in BEAST v. 2.1.3[43]. This method estimates a posterior distribution of effective population sizes through time via MCMC procedures, by moving backward in time until the most recent common ancestor is reached. The constant population size coalescent model was the basic assumption used for this approach. The among-site rate heterogeneity across all branches model, a strict molecular clock and a substitution rate of $0.04 \times 10^{-9}$ /site/year[44]. Markov chains were run for $2.5 \times 10^7$ generations and were sampled every 1000 generations, with the first 2500 samples discarded as burn-in. Other parameters were set as default values and results were visualized in the TRACER v. 1.6 [45].

For the phylogenetic analysis, Bayesian inferences (BI), Maximum Likelihood (ML) and Neighbor-Joining (NJ) trees were reconstructed in MEGA [36] and the BEAST v.2.1.3 [46]. Nodal support was estimated using bootstrap analysis with 1000 replicates. Evolutionary relationships between sequences were assessed also with median-joining networks constructed with the program NETWORK v.4.5.1.0 (www.fluxus-engineering.com/network_terms.htm) using a median-joining algorithm and default settings as in Bandelt et al. (1999)[47]. Further, final phylogenetic tree were constructed using the Bayesian method using the sequences of all wolf sub-species distributed throughout the globe (S1 Table). Phylogenetic analyses were performed for 20 million generations while sampling every 1000th tree, and the first 10% of trees sampled were treated as burn in, and FigTree v.1.3.1 [48]was used to display and summarized annotated phylogenetic trees yielded by BEAST.

## Results

### Sequence variability, sequence divergence and haplotypes

We sequenced total 292 bp of the mitochondrial control region (mtCR) throughout the distribution range of Woolly wolf in the Himalayan region, resulting into an alignment of 263 bp. After merging our data with the sequences retrieved from the GenBank, a total 246 bp long alignment was retained for subsequent analysis. A total of 18 variable sites were observed resulting into 15 haplotypes (Table 1, Fig 1). We found two haplotypes in our data (Table 2). Haplotype 2was shared between the Himachal Pradesh and Uttarakhand, whereas haplotype 3was found in the Jammu and Kashmir, Uttarakhand and Sikkim (Tables 1 and 2; Fig 2). The sequence divergence between the haplotypes ranged from 0.000 to 0.044. The least divergent (0.001) were betweenH14 and H8 whereas highest divergence (0.044) was between H9 and H15 (S2 Table). The inter species sequence divergence was found between the 0.09–0.15 between the canids species (S4 Table). The sequence divergence between the Tibetan wolf and wolf of mainland was 0.09 which is slightly lower (0.13–0.16) than the sequence divergence observed in other canid species.

### Diversity and neutrality indices

The nucleotide ($\pi$) and haplotype (h) diversity among the Indian samples of the Woolly wolf were 0.00270 and 0.588, respectively. These values were relatively higher in PRC ($\pi$ = 0.01278 and h = 0.888) than the Himalayan region including Nepal ($\pi$ = 0.00344 and h = 0.606) (Table 3). The overall nucleotide and haplotype diversity of the Woolly wolf were 0.00851 and 0.777, respectively (Table 3). The two widely used neutrality tests, Tajima's D and Fu's Fs, were found positive, but statistically non-significant for the Indian samples. For all samples, the

**Table 1. Nucleotide variability between different haplotypesobserved from the wild collected samples of Indian Himalayan region and other published sequences of Woolly wolf (*Canis lupus chanco*) using a mitochondrial control region (246 bp).**

| Haplotypes | Nucleotide position | | | | | | | | | | | | | | | | | | |
|---|---|---|---|---|---|---|---|---|---|---|---|---|---|---|---|---|---|---|---|
| | 1 | 1 | 1 | 1 | 1 | 1 | 1 | 1 | 1 | 1 | 1 | 1 | 1 | 1 | 1 | 1 | 1 | 1 | 1 |
| | 5 | 5 | 5 | 5 | 5 | 5 | 5 | 5 | 5 | 5 | 5 | 5 | 5 | 5 | 5 | 5 | 5 | 5 | 5 |
| | 4 | 4 | 4 | 5 | 5 | 5 | 5 | 5 | 5 | 5 | 5 | 5 | 6 | 6 | 6 | 6 | 6 | 6 | 7 |
| | 8 | 8 | 9 | 0 | 1 | 1 | 2 | 4 | 5 | 5 | 5 | 6 | 3 | 3 | 4 | 4 | 5 | 5 | 0 |
| | 1 | 8 | 5 | 5 | 4 | 6 | 6 | 6 | 1 | 4 | 8 | 5 | 3 | 8 | 4 | 6 | 2 | 4 | 0 |
| H1 | G | T | T | G | G | G | T | G | T | G | C | C | C | - | G | A | C | A | G |
| H2 | . | . | . | . | . | . | . | . | . | A | . | . | . | - | A | . | . | G | . |
| H3 | . | . | . | . | A | . | . | . | . | A | . | . | . | - | A | . | . | . | . |
| H4 | . | . | . | . | . | . | . | . | . | A | . | . | . | T | A | . | . | G | . |
| H5 | . | . | . | . | . | A | . | . | . | A | . | . | . | - | A | . | . | G | A |
| H6 | . | . | . | . | . | . | . | . | . | A | T | . | . | T | A | . | . | G | . |
| H7 | A | C | . | . | . | . | . | . | . | A | . | . | . | - | A | . | . | . | . |
| H8 | . | . | . | . | . | . | . | . | . | A | . | . | . | - | A | . | . | . | . |
| H9 | . | . | G | A | . | A | . | C | A | A | . | . | . | - | T | . | . | . | . |
| H10 | . | . | . | . | . | . | . | . | . | A | . | . | . | - | . | . | . | . | . |
| H11 | . | . | . | . | . | . | . | . | . | . | . | . | . | - | . | G | T | . | . |
| H12 | . | . | . | . | A | . | A | . | . | A | . | . | . | - | A | . | . | . | . |
| H13 | . | . | . | . | . | . | . | . | . | A | . | . | . | T | T | . | . | . | . |
| H14 | . | . | . | . | . | . | . | . | . | A | . | . | . | T | A | . | . | . | . |
| H15 | . | . | . | . | . | . | . | . | . | A | . | T | . | T | . | G | T | . | . |

**Table 2. Observed haplotypes in Woolly wolf (*Canis lupus chanco*) across its range in relation to geographical areas.**

| Zones | Geographical regions | No. of samples used | Haplotypes | | | | | | | | | | | | | | | Reference |
|---|---|---|---|---|---|---|---|---|---|---|---|---|---|---|---|---|---|---|
| | | | H1 | H2 | H3 | H4 | H5 | H6 | H7 | H8 | H9 | H10 | H11 | H12 | H13 | H14 | H15 | |
| Himalayan Region | | | | | | | | | | | | | | | | | | |
| | JK, India | 2 | | | 1 | | | | | | | | | | 1** | | | Present study and Sharma et al. 2004 |
| | HP, India | 15 | | 15 | | | | | | | | | | | | | | Present study |
| | HP, India (Zoo) | 17 | | | | | | | | 17*** | | | | | | | | Aggarwal et al. 2007 |
| | UK, India | 2 | | 1 | 1 | | | | | | | | | | | | | Present study |
| | Sikkim, India | 1 | | | 1 | | | | | | | | | | | | | Present study |
| | Nepal | 3 | | | | 1** | | | 1* | | | | | | | 1** | | Chetri et al. 2016 |
| QTP | Tibet | 3 | 1* | | | | | 1** | | | | | | | | | 1** | Meng, 2009 |
| | Unknown location, PRC | 9 | 2* | | 1* | | | | | 3* | 1* | 1* | 1* | | | | | GenBank |
| | Qinghai Lake, PRC | 13 | 2* | | 2* | | 1* | | | 4* | 1* | 1* | 1* | 1* | | | | GenBank |
| Mongolia | Mongolia | 2 | 2* | | | | | | | | | | | | | | | GenBank |

JK = Jammu and Kashmir; HP = Himachal Pradesh; UK = Uttarakhand; China UK = China Unknown locations; China QL = China, Qinghai Lake

PRC = People's Republic of China; QTP = Qinghai Tibetan Plateau

* Observed in earlier studies (published sequences)

** Observed in museum samples

*** Observed in captive samples

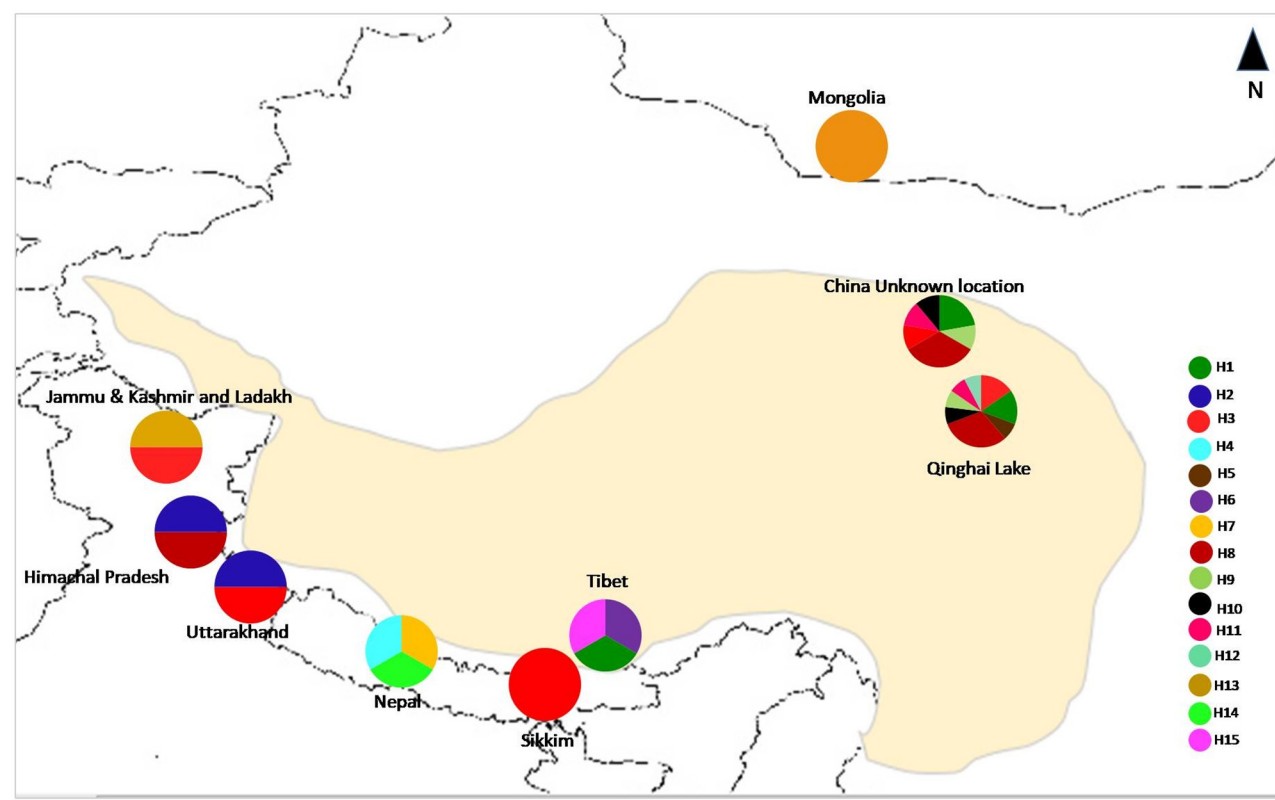

**Fig 2. Haplotype distribution frequency of *Canis lupus chanco* from different geographical regions using the mitochondrial control regions (map not to scale).**

**Table 3. Mitochondrial diversity indices and neutrality tests observed using sequences of the present study and so far reported from other areas for mitochondrial control region (246 bp) in Woolly wolf (*Canis lupus chanco*).**

| Sampling locations | N | Nh | Nucleotide Diversity (π) | Haplotype diversity (h) | Tajima value (D) | Fu' Fs | SSD | r |
|---|---|---|---|---|---|---|---|---|
| India (UK, JK, HP, SK) | 37 | 3 | 0.00270 | 0.588 | 0.73636 | 0.782 | 0.02870 (0.03000) | 0.20666 (0.02000) |
| Himalayan region | 40 | 4 | 0.00344 | 0.606 | -0.49014 | -0.055 | 0.02870 (0.03000) | 0.20666 (0.02000) |
| QTP, PRC | 25 | 11 | 0.01278 | 0.888 | -0.95919 | -1.936 | | |
| Mongolia | 2 | 1 | - | - | - | - | - | - |
| Overall | 66 | 15 | 0.00878 | 0.790 | -1.65226* | -4.175 | 0.00392 (0.69000) | 0.03157 (0.68000) |

N = number of samples, Nh = number of haplotype diversity,

JK = Jammu and Kashmir; HP = Himachal Pradesh; UK = Uttarakhand; QTP = Qinghai Tibetan Plateau; *PRC* = People's Republic of China

*Statistically significant; values in parentheses are significance values

Tajima's D and Fu's Fs statistics values were negative, with only Tajima's D value statistically significant (P<0.05) (Table 3). The mismatch distribution graphs from all the data and from our samples indicate a multimodal pattern (S1 Fig). For all the data, there was a high ruggedness (r) value (0.03157; P = 0.68000) and sum of square deviation (SSD; 0.00392 P = 0.69000), which are statistically non-significant (Table 3). The Bayesian skyline plots show a stable population size over the time except for a recent decline (Fig 3).

## Network and phylogenetic tree analysis

Median-Joining network spanned a total of 15 haplotypes with two median vectors which show the missing connecting haplotypes (Fig 1). Most of the haplotypes are separated with 1–3 mutations, except H 9 in PRC that is separated by five mutations. Bayesian (BI) phylogenetic tree used with HKY model which were second suitable model for the interpretation as all the methods have shown similar topology. In the tree, the haplotypes form six major clades. In most clades, the haplotypes originate from multiple locations (S2 Fig), except in clades 1, 2 and 3. No clear phylogeographic pattern was seen in population genetic differentiation either. In overall, phylogenetic tree of all wolf subspecies splits in two major clade one as Woolly wolf clade and second is wolf-dog clade (Fig 4). These clades indicates that Woolly wolf and wolf-

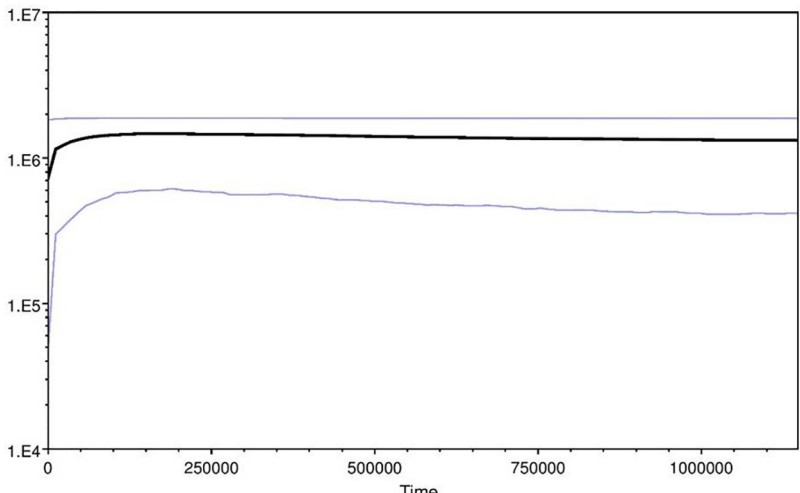

**Fig 3. Bayesian skyline plot of *Canis lupus chanco* using mitochondrial control region.**

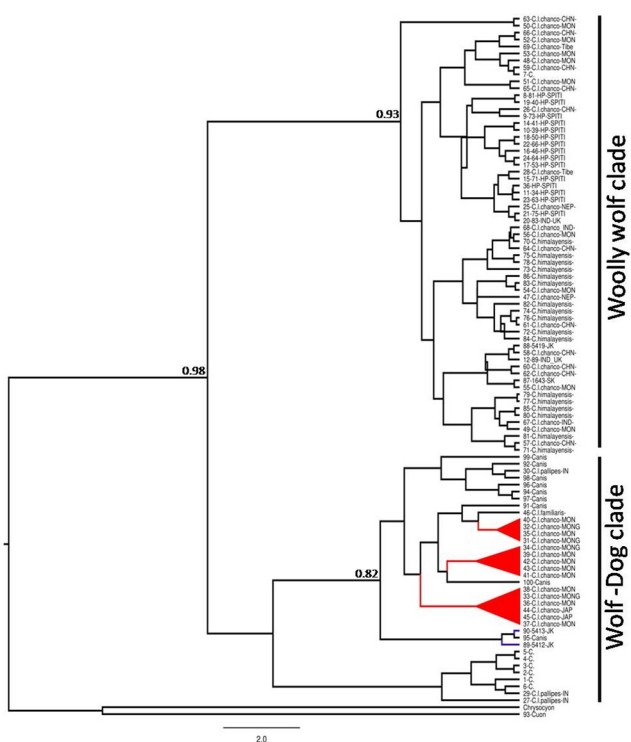

**Fig 4. Overall phylogenetic relationships of Woolly wolf and wolf-dog clade using the mitochondrial control region *Cuon alpinus* and *Chrysocyon brachyurus* are as the out-group.** Clade highlighted in red are the samples marked as *C. l. chanco* [49] and [50]. Blue lines indicates samples collected from the Himalayan range.

dog clades adequately diverged with the mean genetic divergence (0.08 SD). $F_{ST}$ values calculated between the clades ranged from 0.533–1.00 (Table 4), however, they were not significant when calculated between geographic areas and higher values may be artifact of sample size. Significant differences were observed between clade 1 and 2 and between clades 5 and 6 when compared to the other clades (Table 4). The AMOVA, partitioned 71.35% of variation among the populations and 28.65 within the populations (S3 Table), with a highly significant $F_{ST}$ (P = 0.0000; S3 Table).

## Discussion

Phylogenetic or taxon status of the Woolly wolf from the Himalayan region in relation to QTP and Mongolia has been poorly known due to lack of samples from the wild [15,18]except in recent studies from Nepal [4,10,11,19]. By combining sequences from other studies, altogether seven haplotypes have been detected in the Himalayan region (Tables 1 and 2). The overall nucleotide diversity of the Woolly wolf across its range in Central Asia (0.00878;Table 3) was lower than in other wolf species of India and Central Asia (π 0.04-.012; Hd 0.11–0.94)[31] and golden jackal (π 0.0161)[51]. The nucleotide and haplotype diversities in the Woolly wolf of Himalaya were lower than at the QTP region (Table 3). Such higher mtDNA diversity in QTP region has also been reported among other species [52,53].

In corroboration to earlier findings [4,10,11,15,18,19], Woolly wolves from western to eastern Himalayan region were basal and monophyletic in the phylogenetic tree of Holarctic wolf and wolf-dog clades. The present-day monophyletic lineages of the Woolly wolf from Central Asia clustered in six sub-clades (S2 Fig). The Median-Joining (MJ) Network analysis revealed

**Table 4. Estimate of $F_{ST}$ values in the different geographic areas and clades using a mitochondrial control region of Woolly wolf (*Canis lupus chanco*).**

(a) Between geographic areas in Himalayan region

|  | HP | QTP | UK | Nepal | JK | SK |  |
|---|---|---|---|---|---|---|---|
| Himachal Pradesh (HP) |  |  |  |  |  |  |  |
| QTP | 0.74432 |  |  |  |  |  |  |
| Uttarakhand (UK) | 0.77612 | 0.04545 |  |  |  |  |  |
| Nepal | 0.79866 | 0.14286 | 0.7956 |  |  |  |  |
| Jammu and Kashmir (JK) | 0.8783 | -0.11628 | -0.2 | -0.0625 |  |  |  |
| Sikkim (SK) | 1 | -0.2 | -1 | -0.875 | -1 | 0 |  |

(b) Between the clades

|  | Areas | Clade1 | Clade2 | Clade3 | Clade4 | Clade5 | Clade6 |
|---|---|---|---|---|---|---|---|
| Clade1 |  | 0 |  |  |  |  |  |
| Clade2 |  | 0.66667* | 0 |  |  |  |  |
| Clade3 |  | 1.0000 | 0.53398 | 0 |  |  |  |
| Clade4 |  | 1.0000 | 0.81887 | 1.0000 | 0 |  |  |
| Clade5 |  | 0.88664* | 0.87110* | 0.81872* | 0.94174* | 0 |  |
| Clade6 |  | 0.80408* | 0.84528* | 0.63844* | 0.92004* | 0.67279* | 0 |

The $F_{ST}$ values marked with * are significant ($P = 0.05$).

Clade 1: People's Republic of China (PRC); Clade 2 Mongolia; Clade 3: PRC; Clade 4 India and PRC; Clade 5: India and PRC; Clade 6: India, QTP, Nepal and PRC.

H8 and H2 as core haplotypes (Fig 1). The topology in Median-Joining (MJ) Network indicates intermixing of different populations, supported by the low $F_{ST}$-values among regions (Table 4). AMOVA analysis reveals a relatively higher variation between populations than within population (S3 Table). Harp ending's raggedness index, multimodal mismatch distributions, high SSD-value and lack of a star-like Median-Joining Network together with the Bayesian Skyline result support stable long-term population size [54,55]of the Woolly wolf in this area.

A similar trend was also reported in Golden Jackal [51]. The Bayesian Skyline analysis also indicated a decline of the effective population size in the recent time (Fig 3). Such a recent decline has also been reported from the brown bear [56], Tibetan antelope (*Pantholops hodgsonii*) and Tibetan gazelle (*Procraprapicticaudata*) [53] in this landscape. The lack of geographic affinity of haplotypes observed within the sub-clades of *C. l. chanco* (S2 Fig) has also observed at the global scale for wolves from the different parts of Eurasia[10]. The shallow divergence within the Woolly wolf clade indicates that all the populations in Central Asia probably emerged within a very short period from ancestor. Our analyses (S2 Fig; S2 Table) indicate that the ancestor of present-day Woolly wolf lineage probably had ranging behavior covering longer distances across its range during the glaciation-deglaciation periods and the population was quite stable despite of climatic oscillation (Fig 3).

## Presence of regional private haplotypes and paleo-ecological perspective

Even though the Woolly wolf has extremely large territories(6670–26619 km$^2$) in Mongolia [57] and 800–1000 km$^2$ in the Himalayan region (Lyngdoh et al. unpublished), and therefore same haplotypes were found over the large areas, however, some regional geographic affinity was also observed (Table 2; Fig 5). Of the two haplotypes (H2 and H3) collected from the wild along Indian Himalaya, H3 was shared among the locations in Jammu and Kashmir, Uttarakhand, Sikkim, and QTP (Table 2) and may be ancient haplotype. This indicates connectivity of populations through the rugged terrains of southern QTP which is similar to Himalayan region (Fig 5) and may have probably facilitated the movement of the female wolf between

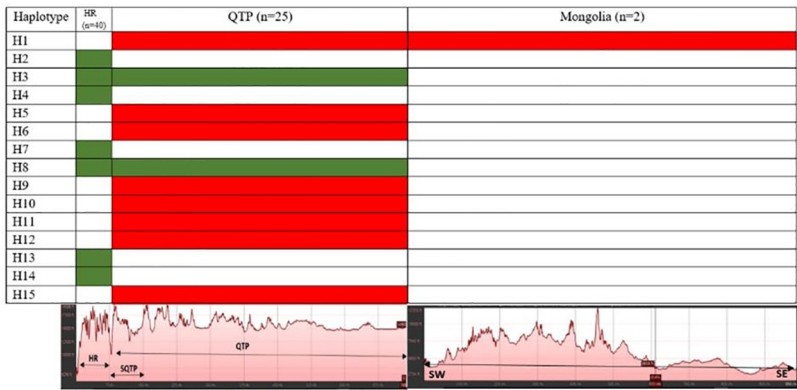

**Fig 5. Distribution of haplotypes in relation to terrain profile of Himalayan region (HR), QTP and Mongolia.**
(SQTP = South QTP; SW = Southwest; SE = Southeast). Red colour represent the haplotype distributed only in the QTP, PRC green represent the haplotype only distributed in the Himalaya and QTP.

western and eastern Himalaya in the past. Presence of similar genetic connectivity between Jammu and Kashmir to Sikkim has also been observed in the snow leopard based on mtDNA and nuclear markers (Singh et al. unpublished). The observed Haplotype H2 in the present study was restricted to Himachal Pradesh (HP) and Uttarakhand. Similarly, in the Himalayan Brown bear which is sympatric with Woolly wolf in this landscape has also shared a haplotype only between HP and Uttarakhand in the western Himalayan region (Goyal et al. unpublished). These observations may indicate the presence of suitable corridor connecting habitats between HP and Uttarakhand, whereas such connectivity may probably lacking with other parts, especially with JK in the northern region. However, this should be validated by monitoring GPS tagged Woolly wolf individual and GIS analysis. Haplotype H7 was also restricted to the wild population of Nepal and has not been reported from anywhere else (Table 2). Absence of QTP haplotypes in Mongolia may be due to Quilon mountains (Fig 2), which have been a barrier for some species after deglaciation periods may be due to differences in availability of the wild prey species across these two regions (Fig 5). Observed private regional haplotypes may indicate multi-stage population divergence in the past(Fig 5), as these areas are well known for differential glacial and interglacial periods across Himalaya and QTP even after the LGM [58,59]. This may have been the reason of different ectomorphs of the Woolly wolf reported at different parts of its range [12]. Similar divergence from multiple refugia has been reported for Himalayan snow cock (*Tetraogallus himalayensis*) in this landscape [60].

Presence of private haplotypes (Table 2; Fig 5) may indicate that there are some restrictions to dispersal. This may indicate significance of some physical, ecological or biological factors that limit the spread of haplotypes to areas, which would be within the dispersal ability of the Woolly wolf[57](Lyngdoh et al unpublished). The time used to search and pursue for prey influences the foraging strategy of a predator and this behavior has been found to differ across terrain complexities and dominate prey species [61]. Wolves occur in varied topographies and differ significantly in the type, density and selection of prey species across the world[62–65]. We also noted that different terrain complexities (Fig 5) and variation in the availability of the wild prey species guilds affect the dietary habits of the Woolly wolf across its range (Fig 5). Medium to large sized species were major prey species in the Himalayan range, whereas small sized species were preferred in QTP and Mongolia (Fig 6).

Genetic structuring have been observed in wolves despite the absence of topographic barriers limiting dispersal. Known drivers for this have been familiarization to natal habitat, prey

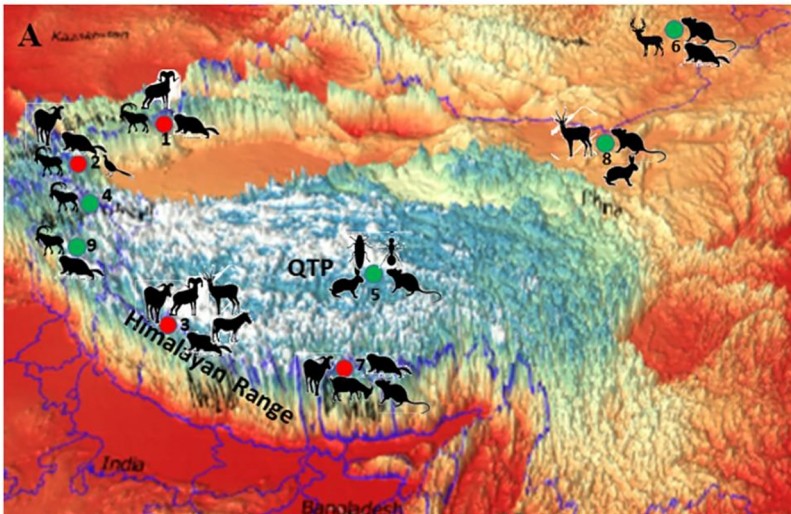

**Fig 6. Variation in terrain complexities and wild prey species in the diet of Woolly wolf based on different studies (1–9) across its range in Central Asia.** (1: [66]; 2: [67]; 3: [68]; 4: [69]; 5: [70]; 6: [71]; 7: [72]; 8: [73]; 9: [74]).

specialization [17,75–78]and probably hunting skills to particular wild prey species that were passed on from parents to their offspring[78]. In the absence of information on species' ranging and foraging behavior and in concurrence with the literature, we speculate that these drivers and the past paleoclimatic conditions over Himalaya-QTP-Mongolia regions might have been the reasons for the observed private haplotypes (Table 2; Fig 6).

The haplotypes which have been detected either in the museum or zoo samples (Table 2), have so far not been reported in wild samples in spite of reasonably extensive sampling in the present study (n = 19) and in Nepal (n = 72)[4,19]. We suggest a need of extensive sampling in the Himalayan region to find for the haplotypes reported from museums or zoos in the wild. Moreover, such sampling strategies may shed light on existing barriers of gene flow in the Himalayas.

## Sequence ambiguity in Woolly wolf: Are they Himalayan or Tibetan or Mongolian?

Earlier study indicated that the Himalayan wolf is a separate basal clade among the other wolves of the world[18]. In their analysis, samples of Grey wolf from peninsular India (*C. l.pallipes*) and a total of five sequences of *C. l. chanco* originating from Mongolia and also used in a another study [49]clustered in the same clade, whereas Himalayan wolf sequences generated by them were in a separate ancient lineage. Therefore, it was suggested that the Himalayan wolf should be considered as a separate species *C. himalayensis*[18]. However, sequences of the present study along with all available sequences of the *C. l. chanco* and *C. himalayensis* clustered in a single Woolly wolf clade whereas sequences of *C. l. chanco* used in another study[49] were clustered in Dog-wolf clade (Fig 4). We, therefore, believe that these sequences may not be of the Woolly wolf. Such ambiguity in the sequences has also been reported in other related canids [29,79]. When these sequences were removed, we found no evidence for *Canis lupus chanco* to be distinct from the Himalayan wolf. Therefore, we advocate that there is only one extant subspecies *C. l. chanco* throughout the Himalayan and Central Asian Highlands (Fig 4).

### A need for classifying Woolly wolf as separate conservation unit

Available evidence also suggests the presence of genetic and ecological adaptations that are unique to Woolly wolf across Himalaya, QTP, and Mongolia compared to other wolves. We believe that positive selection of hypoxia-related genetic selection [80] and the placement of all the studied populations of Himalaya, QTP and Mongolia in the same distinct ancient lineage, *C.l. chanco* ~ *C.l. laniger* ~ *C.lhimalayensis* need special conservation and recognition as a separate Evolutionary Significant Unit (ESU). Due to the presence of private regional haplotypes and ecological specialization, as a result from differences in terrain characteristics and differences in the availability of prey species, we advocate further studies using different genetic markers to identify Management Units [81] or Evolutionary Significant Units across range of this sub-species. These conservation units would form a basis for planning effective conservation strategies and would provide scientific support for retaining the regional level genetic integrity and avoid any mixing of gene pool among populations of different ecological specialization. Therefore, we suggest a need of trans-boundary studies using harmonized multi genetic markers so as to ascertain whether Woolly wolf may be designated as a separate species i.e. *Canischanco* or not. "Chanco" is a latinised pronunciation for wolf among the Tibeto-Mongoloid community in Central Asia, encompassing a large proportion of the Woolly wolf range. A recent study on Woolly wolf from Nepal, based on different molecular markers also suggested considering it as a distinct species [4,19].

## Conclusion

Woolly wolf samples from the wild from the western to eastern Himalayan region along with other samples from Nepal, QTP, and Mongolia were placed within the same basal clade in phylogenetic analyses reconfirming that the Woolly wolf clade is ancient to wolf and wolf-dog clades. Because of shallow divergence, non-significant $F_{ST}$ values and low genetic distances among the sub-clades within the monophyletic clade, our study suggest that Woolly wolf is, in fact, *Canis lupus chanco* and not *Canishimalyansis or C.l.laniger*. We observed lower genetic diversity in the populations of the Himalayan region than in QTP. Our analysis reveals that the Woolly wolf deserved recognition as an ESU due to its distinct evolution from paleo-ecological times, adaptation to hypoxic conditions, terrain complexities and unique wild prey assemblage in comparison to other Holarctic wolf populations[64,82]. Furthermore, wolves are known to range longer distances, therefore, there is also a need for trans-boundary efforts in research and conservation management for (i) identifying critical habitats and connectivity corridors through harmonized methods and genetic markers [19,28] in order to ensure long-term survival of this ancient lineage and (ii) use of harmonized multi genetic markers to ascertain whether the Woolly wolf may be designated as separate species or not. Besides the area has the long-standing history of persecution of wolves that is still prevalent. Therefore, a need for effective protection to minimize the wolf-human conflict due to livestock depredation is needed for ensuring viable population for effective conservation in this landscape.

## Supporting information

**S1 Table. Details of samples and Sequences used in this study.**
(DOCX)

**S2 Table. Sequence divergence between the different haplotypes (H1-H15) of Woolly wolf (*Canis lupus chanco)* using a mitochondrial control region.**
(DOCX)

**S3 Table. AMOVA results of Woolly wolf in phylogenetic tree of different clades.**
(DOCX)

**S4 Table. Inter species genetic distance of the family canidae using control regions marker.**
(DOCX)

**S1 Fig. Pairwise mismatch distribution graph of *Canis lupus chanco* using a mitochondrial control region.**
(DOCX)

**S2 Fig. Phylogenetic analysis based on Bayesian inferences (BI) tree constructed of *Canis lupus chanco* using mitochondrial control region of *Cuon alpines* and *Chrysocyon brachyurus* as the out group.** Map showing the clade wise distribution of samples in their distribution range. Values above the nodes are posterior probabilities.
(DOCX)

## Acknowledgments

The authors are grateful to the Director, Dean and Research Coordinator, Wildlife Institute of India, Dehradun, for their strong support and facilitation. Our sincere thanks are due to the Forest Department of different states for providing necessary permission and support during the field and sample collection. The authors acknowledge the support provided by the Nodal Officer and all the researchers and staff of the Wildlife Forensic Cell of the Wildlife Institute of India for conducting this study.

## Author Contributions

**Conceptualization:** BheemDutt Joshi, Surendra Prakash Goyal.

**Data curation:** BheemDutt Joshi, Salvador Lyngdoh, Ved Prakash Tiwari, S. A. Dar, Aishwarya Maheswari, Ranjana Pal, Tawqir Bashir, Hussain Saifee Reshamwala, Shivam Shrotriya, S. Sathyakumar.

**Formal analysis:** BheemDutt Joshi, Vinay Kumar.

**Funding acquisition:** Surendra Prakash Goyal.

**Investigation:** Surendra Prakash Goyal.

**Methodology:** BheemDutt Joshi.

**Project administration:** S. Sathyakumar.

**Software:** BheemDutt Joshi, Shivam Shrotriya.

**Supervision:** Bilal Habib.

**Writing – original draft:** BheemDutt Joshi.

**Writing – review & editing:** BheemDutt Joshi, Salvador Lyngdoh, Sujeet Kumar Singh, Reeta Sharma, Vinay Kumar, Ved Prakash Tiwari, Aishwarya Maheswari, Bilal Habib, Laura Kvist, Surendra Prakash Goyal.

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
