## [Decision Letter · Decision Letter 0]

5 Feb 2020

PONE-D-20-00939

Revisiting the Woolly wolf (Canis lupus chanco) phylogeny in Himalaya: Addressing taxonomy, spatial extent and distribution of an ancient lineage in Asia

PLOS ONE

Dear Dr. Goyal,

Thank you for submitting your manuscript to PLOS ONE. After careful consideration, we feel that it has merit but does not fully meet PLOS ONE’s publication criteria as it currently stands. Therefore, we invite you to submit a revised version of the manuscript that addresses the points raised during the review process.

Please address the minor issues. There is no need to produce more data, but rephrase the species status accordingly. The data are not sufficient to draw definite conclusions at this point.

We would appreciate receiving your revised manuscript by Mar 21 2020 11:59PM. To enhance the reproducibility of your results, we recommend that if applicable you deposit your laboratory protocols in protocols.io, where a protocol can be assigned its own identifier (DOI) such that it can be cited independently in the future. For instructions see: http://journals.plos.org/plosone/s/submission-guidelines#loc-laboratory-protocols

We look forward to receiving your revised manuscript.

Kind regards,

Axel Janke

Academic Editor

PLOS ONE

Journal Requirements:

2. In your Methods section, please provide additional location information of the study area, including geographic coordinates for the data set if available.

3. We note that Figures 1, 2 and 5 in your submission contain maps and satellite images which may be copyrighted. All PLOS content is published under the Creative Commons Attribution License (CC BY 4.0), which means that the manuscript, images, and Supporting Information files will be freely available online, and any third party is permitted to access, download, copy, distribute, and use these materials in any way, even commercially, with proper attribution. For these reasons, we cannot publish previously copyrighted maps or satellite images created using proprietary data, such as Google software (Google Maps, Street View, and Earth). For more information, see our copyright guidelines: http://journals.plos.org/plosone/s/licenses-and-copyright.

You may seek permission from the original copyright holder of Figures 1, 2 and 5 to publish the content specifically under the CC BY 4.0 license. 

If you are unable to obtain permission from the original copyright holder to publish these figures under the CC BY 4.0 license or if the copyright holder’s requirements are incompatible with the CC BY 4.0 license, please either i) remove the figure or ii) supply a replacement figure that complies with the CC BY 4.0 license. Please check copyright information on all replacement figures and update the figure caption with source information. If applicable, please specify in the figure caption text when a figure is similar but not identical to the original image and is therefore for illustrative purposes only.

Reviewers' comments:

Reviewer's Responses to Questions

**Comments to the Author**

1. Is the manuscript technically sound, and do the data support the conclusions?

Reviewer #1: Partly

Reviewer #2: Yes

2. Has the statistical analysis been performed appropriately and rigorously? 

Reviewer #1: Yes

Reviewer #2: Yes

3. Have the authors made all data underlying the findings in their manuscript fully available?

Reviewer #1: Yes

Reviewer #2: Yes

4. Is the manuscript presented in an intelligible fashion and written in standard English?

Reviewer #1: No

Reviewer #2: Yes

5. Review Comments to the Author

Reviewer #1: The authors investigated the phylogenetic status of the woolly wolf, a type of wolf that inhabits the Himalayan Region, the Qinghai Tibetan Plateau (QTP) and Mongolia. The authors sequenced the mt control regions of 19 individuals from the Himalayan Region, namely from Jammu and Kashmir (JK, n=1), Himachal Pradesh (HP, n=15), Uttarakhand (UK, n=2) and Sikkim (n=1)). They compared these sequences to 48 wolf mt control region sequences downloaded from Genbank, originating from the Himalayan Region (21), the QTP (Tibet, Qinghai Lake, and unknown location, n = 25) and Mongolia (2). There objectives were twofold: to study population structuring of the woolly wolf, and to reexamine the taxonomic status of the woolly wolf, as previous studies have suggested that the status of the woolly wolf should be elevated to the species level.

In my opinion the authors meet the first objective satisfactorily, but don’t provide enough new data or analyses to meet the second objective. They argue that Himalayan, QTP and Mongolian wolves group together in a basal wolf lineage, and that in addition these wolves are uniquely adapted to height. On these two grounds the authors argue that the woolly wolf should be considered as a separate species.

Although I don’t necessarily disagree, I think more new data and more solid evidence is needed to justify this claim. Since the taxonomic status of the woolly wolf is not the main theme of the paper, the authors could consider downplaying these statements. Alternatively, the authors could expand in the discussion on the species concept discussion (if they really want to go down that road), and discuss how the observed genetic distance of woolly wolves and other wolves compares to the genetic divergence of other carnivore subspecies/species pairs. In addition, figure S3 should became a main figure, and be discussed more thoroughly (both in the methods and the results section).

Considering the presentation of the data, the high number of location names causes a bit of confusion for the reader, at least for me. For the figures, I suggest to use colour coding and/or labelling to clarify whether sampling locations belong to either of the three/four main geographical regions (i.e. India/Himalaya, Qinghai Tibetan Plateau and Mongolia). Additionally, figures could be shrinked and presented as a multi tile plot to facilitate comparisons.

As a final point: although the text is clearly written and easy to follow, there are typo’s and grammar errors (such as the wrong use of ‘also’ and ‘the’) which need to be improved prior to publication. I listed some typo's below, but just a random selection. Please ensure that during revision all typo's and grammatical errors (also the ones not listed below) are corrected.

Title: what do you mean with 'spatial extent'?

Line 41: most studies agree, that… No comma

Line 43: a separate species by some… Remove ‘by some’

Line 52: By combining our data to…. After combining our data with

Line 54: the number of observed haplotypes depends on sample size. Why not use another, sample size independent, estimate for genetic diversity?

Line 56: shallow divergence (0.000-0.044). which estimator?

Line 58: over a long time. How long is long?

Line 58: over the a long…. Remove ‘the’

Line 61: within the range of the Woolly wolf’s…. Remove ‘s

Line 63-65: adaptation is in itself not sufficient to validate species status. As an example: the arctic wolf is uniquely adapted to the polar habitat, but is considered a subspecies and not a species, and many more examples can be given.

Line 67: there is an urgent need of transboundary efforts in identifying management units (MUs) to help conservation. How does this follow from the results?

Line 76: what do you mean with long-ranging mammals?

Line 134-136: I think the objectives could be stated a bit more precisely. I wonder how the objective stated in lines 134_36 differ from re-examining the taxonomy of the woolly wolf, which you already mentioned in the previous sentence? For phylogeography you have not much data to work on. Maybe it is more correct to say that you study the population structure of the Himalayan wolf?

Line 161: and size too?

Line 163: remove ‘a’

Line169: ‘is the highly variable’

Line 181-182: Jammu and Kashmir (n=1), Himachal Pradesh (n=15), Uttarakhand (n=2) and Sikkim (n=1). Does this uneven distribution somehow affect the study outcomes?

Line 193-196: Specify in main text and in abstract how many additional samples. I inferred from the tables that the number is 48 if I didn’t make a calculation error, but this information should be more easy to find.

Line 203: ruggedness indices

Line 217: For the phylogenetic analysis… start new paragraph.

Line: 255 why Fst values so high?

Line 250-252. Only Bayesian (BI) phylogenetic tree used for the interpretation as all the methods have shown similar topology. This explains why you chose so show one tree building method, but not why you choose Bayesian method in particular.

Line 365: Why is figure S3 not mentioned in the results section?

TABLES AND FIGURES

Table 1. what is the meaning of PRC? (stand-alone test: can you understand the figure without reading the main text?)

Table 2: concerning the captive samples, inbreeding could affect the result and is maybe worth a few lines of discussion in either the methods or the discussion section

Figure 1. Could you make the labels a bit bigger? Also, since in table 1 you refer to Himachal Pradesh, for consistency I suggest to replace on the map the text Spiti Valley with Himachal Pradesh or HP. It took me a long time to work out the location of HP samples. Also, the number of colours is quite overwhelming. As I suggested above, it might make it easier for the reader if you use colour coding based on the main geographical regions.

Figure2. I would suggest to make the piecharts a bit bigger, and to crop the map (currently no samples in right third and upper part of the map). Would be even better if the size of the piecharts would be relative to the number of samples.

Figure 4. What do the colours red and green represent?

Figure S1. Is the deviation between expected and observed mismatch distribution significant?

Reviewer #2: This is a well-written and easy to follow manuscript which presents some new results and insight into the taxonomy, phylogeography and conservation genetics of the wolly wolf. The paper has used a non-invasive sampling approach, by collecting faeces in the Field, and also include previously published data/sequences from other geographic areas. Therefore, the results are representative for most of the Himalayan region. I find the study useful, and I only have a few suggestions/questions.

Astract: ok, but is perhaps unncessesarily long? I would recommend to shorten it .

Main text: All parts are well written and relatively easy to follow, at least if you have a background in genetics. However, the Data analysis section in the Methods might be hard to read if you dont have a specialist background. I would recommend to add a few sentences to explain/describe in words what purpose the different statistical analyses have in your study (and perhaps even some expectations/predictions), so that a wider readership can follow and understand the results, e.g. non-geneticist interested in predators and/or Conservation issues. If neccesary, to the same aim, follow up and add a few half sentences in the results too, so that everything connects nicely.

Other than that I find the study well written and interesting.

6. PLOS authors have the option to publish the peer review history of their article (what does this mean?). If published, this will include your full peer review and any attached files.

Reviewer #1: No

Reviewer #2: No

---

## [Author Response · Author response to Decision Letter 0]

21 Mar 2020

Point to point authors response to the Reviewers’ comments

Manuscript ID: PONE-D-20-00939

Manuscript title: Joshi et al. -Revisiting the Woolly wolf (Canis lupus chanco) phylogeny in Himalaya: Addressing taxonomy, spatial extent and distribution of an ancient lineage in Asia

The authors would like express thanks to the reviewer for considering our manuscript interesting and accepting it for publication after revision. We have incorporated all suggestions raised by the reviewer’s for each of the questions in preparation of revised version of manuscript. 

Reviewer #1:

Comments:The authors investigated the phylogenetic status of the woolly wolf, a type of wolf that inhabits the Himalayan Region, the Qinghai Tibetan Plateau (QTP) and Mongolia. The authors sequenced the mt control regions of 19 individuals from the Himalayan Region, namely from Jammu and Kashmir (JK, n=1), Himachal Pradesh (HP, n=15), Uttarakhand (UK, n=2) and Sikkim (n=1)). They compared these sequences to 48 wolf mt control region sequences downloaded from Genbank, originating from the Himalayan Region (21), the QTP (Tibet, Qinghai Lake, and unknown location, n = 25) and Mongolia (2). There objectives were twofold: to study population structuring of the woolly wolf, and to reexamine the taxonomic status of the woolly wolf, as previous studies have suggested that the status of the woolly wolf should be elevated to the species level.

Author response: We appreciate reviewer to evaluate the manuscript carefully. Earlier studies are in support of present study but lack of comprehensive data across Indian Himalayan range previous studies (Aggarwal et al. 2007) suggested that wolf found in the Himalayan region should be designated as separate species. The present study generates largest available samples size of wild Woolly wolf across Himalayas and analysis suggests that Woolly wolf distributed across Mongolia, QTP and India is a single sub-species. We claim that the sub-species may be considered as an ESU though existing studies support the view that the species deserves recognition as a Taxon. 

Comments: In my opinion the authors meet the first objective satisfactorily, but don’t provide enough new data or analyses to meet the second objective. They argue that Himalayan, QTP and Mongolian wolves group together in a basal wolf lineage, and that in addition these wolves are uniquely adapted to height. On these two grounds the authors argue that the woolly wolf should be considered as a separate species.

Author response: We agree with reviewer and have rephrased the species status accordingly (line no. 28-30). However, we suggest that the Woolly wolf may be uplifted to species level from the grey wolf found in the main land of Indian, Pakistan and other regions. 

Comments: Although I don’t necessarily disagree, I think more new data and more solid evidence is needed to justify this claim. Since the taxonomic status of the woolly wolf is not the main theme of the paper, the authors could consider downplaying these statements. Alternatively, the authors could expand in the discussion on the species concept discussion (if they really want to go down that road), and discuss how the observed genetic distance of woolly wolves and other wolves compares to the genetic divergence of other carnivore subspecies/species pairs. In addition, figure S3 should became a main figure, and be discussed more thoroughly (both in the methods and the results section).

Author response: We have downplayed our statements and compared genetic distances within the canids (line no 276-279) and found sequence divergence between the subspecies from 0.03-0.06 of dog wolf clade. Whereas, sequences of dog-wolf clade showed the higher sequence divergence with wooly wolf. Inter species sequences divergence of different canid species were found 0.135-0.163 which is slightly higher than the observed (0.078-090) between the wooly wolf– (Canis lupus chanco) and Grey wolf (Canis lupus lupus/ Canis lupus lupus/ Canis lupus lupus). We brought the figure in main text as Figure 4 and discussed this appropriately both in methods and results section. 

Comments: Considering the presentation of the data, the high number of location names causes a bit of confusion for the reader, at least for me. For the figures, I suggest to use color coding and/or labelling to clarify whether sampling locations belong to either of the three/four main geographical regions (i.e. India/Himalaya, Qinghai Tibetan Plateau and Mongolia). Additionally, figures could be shrinked and presented as a multi tile plot to facilitate comparisons.

Author response: We have changed all suggested figures to readable and increased the size (Figure 1 and Figure 2). 

As a final point: although the text is clearly written and easy to follow, there are typo’s and grammar errors (such as the wrong use of ‘also’ and ‘the’) which need to be improved prior to publication. I listed some typo's below, but just a random selection. Please ensure that during revision all typo's and grammatical errors (also the ones not listed below) are corrected.

Author response: Agreed and manuscript has carefully checked.

Title: what do you mean with 'spatial extent'?

Author response: Spatial extent is geographical extent of the sample coverage

Line 41: most studies agree, that… No comma

Author response: Corrected as per suggestion

Line 43: a separate species by some… Remove ‘by some’

Author response: Corrected as per suggestion

Line 52: By combining our data to…. After combining our data with

Author response: Corrected as per suggestion

Line 54: the number of observed haplotypes depends on sample size. Why not use another, sample size independent, estimate for genetic diversity?

Author response: Corrected as per suggestion

Line 56: shallow divergence (0.000-0.044). Which estimator?

Author response: Corrected as per suggestion and provided in details of all parameters.

Line 58: over a long time. How long is long?

Author response: Corrected as per suggestion

Line 58: over the a long…. Remove ‘the’

Author response: Corrected as per suggestion

Line 61: within the range of the Woolly wolf’s…. Remove ‘s

Author response: Corrected as per suggestion

Line 63-65: adaptation is in itself not sufficient to validate species status. As an example: the arctic wolf is uniquely adapted to the polar habitat, but is considered a subspecies and not a species, and many more examples can be given.

Author response: Apart from adaptation, there are other factors that can be argued in this context, i.e isolation in a large area and time, in the line of African Wolf. Also there are morphological traits that are distinct and not found in many closely related sister species. Even their howls are distinct (Hennelly et al 2017). So there are various other studies which support this finding (Werhan et al 2020). There are also dietary factors which make this wolf uniquely adapted to preying on high altitude species (Lyngdoh et al 2020). There are different species concepts that can fit this justification however, as agreed with the reviewer we advocate a single species in the entire region distinct from the Holarctic wolf and do not recognize Himalayan Wolf as separate from Tibetan Wolf. There lines have modified now

Line 67: there is an urgent need of transboundary efforts in identifying management units (MUs) to help conservation. How does this follow from the results?

Author response: This follows from the recognition of the species as a single entity distinct from Holarctic wolves. The Woolly wolves are well adapted ecological, biologically and morphologically. From available literature, it seems wolves range long distances and often cross political boundaries. In trans-Himalayas and Tibetan Region, prey populations are depressed (Lyngdoh et al 2019). These regions support wolves which may be in huge conflict due to practices that prevail and legislative mechanisms in countries. Therefore, following our data and field data, we know that wolves are distinct yet long ranging, hence trans-boundary conservation efforts are essential for the natural gene flow amongst these individuals. 

Line 76: what do you mean with long-ranging mammals?

Author response: The species has large home range (> 1000 sq. km) across trans-boundary and displays altitudinal migratory behavior from field data behviour such as Tibetan wolf have recorded as larger home ranges (6670–26619 km2 in Mongolia (Kaczensky et al., 2008) and 800–1000 km2 in the Himalayan region (Lyngdoh et al. unpublished). 

Line 134-136: I think the objectives could be stated a bit more precisely. I wonder how the objective stated in lines 134_36 differ from re-examining the taxonomy of the woolly wolf, which you already mentioned in the previous sentence? For phylogeography you have not much data to work on. Maybe it is more correct to say that you study the population structure of the Himalayan wolf?

Author response: Necessary changes made as suggested in line number 124-128.

Line 161: and size too?

Author response: Corrected as per suggestion

Line 163: remove ‘a’

Author response: Corrected as per suggestion

Line169: ‘is the highly variable’

Author response: Corrected as per suggestion

Line 181-182: Jammu and Kashmir (n=1), Himachal Pradesh (n=15), Uttarakhand (n=2) and Sikkim (n=1). Does this uneven distribution somehow affect the study outcomes?

Author response: No these may lead to increase of some haplotype but will not change overall assignment and population structure

Line 193-196: Specify in main text and in abstract how many additional samples. I inferred from the tables that the number is 48 if I didn’t make a calculation error, but this information should be more easy to find.

Author response: All the details have been given in text in the line no. 213

Line 203: ruggedness indices

Author response: Corrected as per suggestion

Line 217: For the phylogenetic analysis… start new paragraph.

Author response: Corrected as per suggestion

Line: 255 why Fst values so high?

Author response: This may be due to small samples size 

Line 250-252. Only Bayesian (BI) phylogenetic tree used for the interpretation as all the methods have shown similar topology. This explains why you chose so show one tree building method, but not why you choose Bayesian method in particular.

Author response: As Bayesian tree widely used and also the best suitable nucleotide substitution modes HKY were available with BEAST package and reasons have been provided in the methods in details. 

Line 365: Why is figure S3 not mentioned in the results section?

Author response: As suggested, this has been brought as main figure and discussed accordingly in line no 302-314.

TABLES AND FIGURES

Table 1. what is the meaning of PRC? (stand-alone test: can you understand the figure without reading the main text?)

Author response: All abbreviation have been provided carefully and clarity of text have been given

Table 2: concerning the captive samples, inbreeding could affect the result and is maybe worth a few lines of discussion in either the methods or the discussion section

Author response: Use of maternal inherited genetic markers is not affected by inbreeding because the respective haplotype would be present in next progeny. Therefore, inclusion of captive samples in our analysis may not affect our conclusion.

Figure 1. Could you make the labels a bit bigger? Also, since in table 1 you refer to Himachal Pradesh, for consistency I suggest to replace on the map the text Spiti Valley with Himachal Pradesh or HP. It took me a long time to work out the location of HP samples. Also, the number of colours is quite overwhelming. As I suggested above, it might make it easier for the reader if you use colour coding based on the main geographical regions.

Author response: Yes Agreed

Author response: We have made appropriate changes 1) Increased the labels size, changed spiti valley to Himachal Pradesh and subsequent in the text. In this figure 1 we would like show the individually haplotypes of wild and museum/zoo samples therefore the haplotype coloring pattern is given in map as per Haplotype Network. However we have changed size and labels for the clarity. 

Figure 2. I would suggest to make the piecharts a bit bigger, and to crop the map (currently no samples in right third and upper part of the map). Would be even better if the size of the piec harts would be relative to the number of samples.

Author response: Necessary changes were made in Figure.

Figure 4. What do the colours red and green represent?

Author response: Labels have been provided.

Figure S1. Is the deviation between expected and observed mismatch distribution significant?

Author response: No there is no significant deviation.

Reviewer #2: This is a well-written and easy to follow manuscript which presents some new results and insight into the taxonomy, phylogeography and conservation genetics of the Woolly wolf. The paper has used a non-invasive sampling approach, by collecting faeces in the Field, and also include previously published data/sequences from other geographic areas. Therefore, the results are representative for most of the Himalayan region. I find the study useful, and I only have a few suggestions/questions.

Author response: We thank reviewer for positive response. 

Astract: ok, but is perhaps unncessesarily long? I would recommend to shorten it .

Author response: The abstract is reduced as suggested

Main text: All parts are well written and relatively easy to follow, at least if you have a background in genetics. However, the Data analysis section in the methods might be hard to read if you do not have a specialist background. I would recommend to add a few sentences to explain/describe in words what purpose the different statistical analyses have in your study (and perhaps even some expectations/predictions), so that a wider readership can follow and understand the results, e.g. non-geneticist interested in predators and/or Conservation issues. If necessary, to the same aim, follow up and add a few half sentences in the results too, so that everything connects nicely.

Author response: All required suggestion has been incorporated.

Other than that I find the study well written and interesting.

Author response: We thanks to reviewers for reviewing the manuscript from line 223-237.

---

## [Editor Report · Decision Letter 1]

30 Mar 2020

Revisiting the Woolly wolf (Canis lupus chanco) phylogeny in Himalaya: Addressing taxonomy, spatial extent and distribution of an ancient lineage in Asia

PONE-D-20-00939R1

Dear Dr. Goyal,

We are pleased to inform you that your manuscript has been judged scientifically suitable for publication and will be formally accepted for publication once it complies with all outstanding technical requirements.

Please follow my instructions in the comments to the authors. They are not mandatory,  but in your own interest to improve the value of the fascinating study.

With kind regards,

Axel Janke

Academic Editor

PLOS ONE

Additional Editor Comments (optional):

Dear Authors

1) the abstract is not shortened as suggested by one reviewer. I opted to accept the paper, because I _trust_ that you will shorten it to a standard size. Abstracts typically have 200-300 Words. It almost 500 words now!

2) I also trust that you go carefully through the MS again and implement and add a few explanations that you provided to reviewer's one questions to improve the MS, e.g. what is a long-ranging species, what you you mean by spatial and so on. While you have dutifully answered the questions in the reply, some of that did not make it into the MS. This is not crucial and at your discretion, but it is in your own interest.
---

## [Editor Report · Acceptance letter]

3 Apr 2020

PONE-D-20-00939R1 

Revisiting the Woolly wolf (*Canis lupus chanco*) phylogeny in Himalaya: Addressing taxonomy, spatial extent and distribution of an ancient lineage in Asia 

Dear Dr. Goyal:

I am pleased to inform you that your manuscript has been deemed suitable for publication in PLOS ONE. Congratulations! Your manuscript is now with our production department. 

With kind regards,

on behalf of

Dr. Axel Janke 

Academic Editor

PLOS ONE